# Vegas: Self-Speculative Decoding with Verification-Guided Sparse Attention

Yikang Yue [1]   Yuqi Xue [1]   Jian Huang [1]

## Abstract

Long-context large language model (LLM) inference has become the norm for today's AI applications. However, it is severely bottlenecked by the increasing memory demands of its KV cache. Previous works have shown that self-speculative decoding with sparse attention, where tokens are drafted using a subset of the KV cache and verified in parallel against the full KV cache, speeds up inference in a lossless manner. However, they rely on a standalone KV selection algorithm to select the KV entries used for drafting and overlook the fact that the criticality of each KV entry is inherently computed during verification.

In this paper, we propose Vegas, a self-speculative decoding method with verification-guided sparse attention. Vegas identifies critical KV cache entries as a byproduct of verification and computes attention only over these entries when drafting subsequent tokens. This not only improves the draft token acceptance rate but also incurs low KV selection overhead, thereby improving decoding throughput. Vegas achieves a $1.25\times$–$2.81\times$ speedup in decoding throughput over default vLLM and a $1.15\times$–$1.29\times$ speedup over state-of-the-art sparse attention-based self-speculative decoding methods. Our code is available at https://github.com/platformxlab/vegas.

## 1. Introduction

With recent advances in reasoning large language models (LLMs), long-context LLM inference has become the norm (OpenAI, 2025a; Google DeepMind, 2025; Guo et al., 2025; Yang et al., 2025a). However, the long-context LLM inference suffers from memory bandwidth bottlenecks due to its intensive KV cache accesses, as the KV cache size grows linearly with the context length during auto-regressive decoding (Liu et al., 2025b).

To mitigate the memory bottleneck, sparse attention techniques have been proposed to reduce memory access by processing only a subset of critical KV entries (Zhang et al., 2023; Ge et al., 2024). While effective at reducing memory access overhead, these methods inherently risk degrading output quality because they discard portions of the context during generation (Li et al., 2025a; Zhang et al., 2025b).

Self-speculative decoding has emerged as a solution to ensure output quality while exploiting the performance benefits of sparse attention (Sun et al., 2024; Sadhukhan et al., 2025). As shown in Figure 1a, in each decoding iteration, the original model employs sparse attention to *draft* a sequence of tokens auto-regressively, which are then *verified* in parallel in a single forward pass using full attention. Since the full KV cache is accessed only once in the verification stage, the overhead can be amortized across multiple accepted draft tokens to achieve a significant speedup.

The performance of sparse attention-based self-speculative decoding is heavily impacted by the inherent trade-off between drafting accuracy (the percentage of draft tokens accepted) and KV selection overhead (the extra computation required for identifying critical KV entries for sparse attention). Existing works have employed simple sliding window attention with low KV selection overhead (Sadhukhan et al., 2025) or a more sophisticated query-aware sparse attention algorithm to improve drafting accuracy (Sun et al., 2024). They either suffer from low drafting accuracy or incur high KV selection overhead that undermines the speedup of speculative decoding. The fundamental limitation is that these approaches treat drafting and verification as standalone processes (Figure 1a): they utilize the verification phase only to accept or reject tokens. They overlook a critical opportunity: the verification phase, by performing full attention, inherently calculates the exact criticality of every KV entry – a "free oracle" that can be used to select the KV entries for the sparse attention in the drafting phase.

In this paper, we develop Vegas, a self-speculative decoding mechanism with verification-guided sparse attention (Figure 1b). Vegas co-designs the drafting and verification phases: it utilizes the intermediate results of the full attention computed during verification to identify critical KV entries, which then guide the sparse attention computation in the subsequent drafting phase. Since the KV entries are

---

[1]University of Illinois Urbana-Champaign. Correspondence to: Jian Huang <jianh@illinois.edu>.

*Proceedings of the $43^{rd}$ International Conference on Machine Learning*, Seoul, South Korea. PMLR 306, 2026. Copyright 2026 by the author(s).

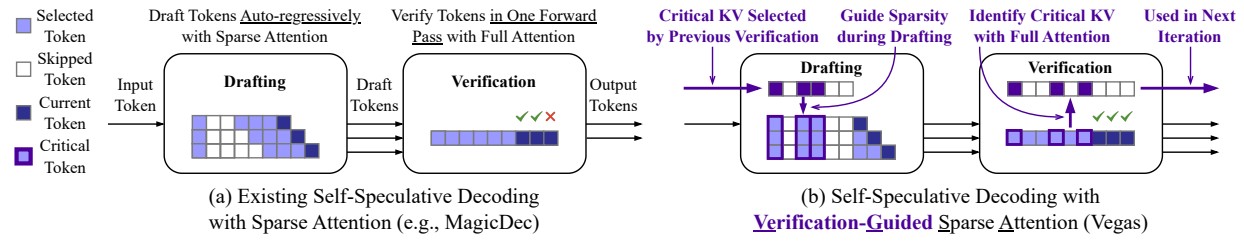

*Figure 1.* Comparison of (a) existing self-speculative decoding with sparse attention and (b) self-speculative decoding with *verification-guided* sparse attention (Vegas). Each subfigure shows one decoding iteration (drafting + verification). Vegas improves decoding throughput by identifying critical KV entries as a byproduct of verification and applying sparse attention over these entries when drafting subsequent tokens, enabling a high draft token acceptance rate. Note that details such as the bonus token are omitted for simplicity.

selected based on full attention, Vegas can achieve high drafting accuracy with low performance overhead.

To realize this co-design efficiently, Vegas addresses two critical challenges: optimizing the KV selection strategy for accuracy and minimizing the runtime overhead of attention logit collection. First, selecting KV entries based solely on the last accepted token leads to overfitting, causing accuracy to decay rapidly for subsequent draft tokens. To address this, we select KV entries by aggregating attention logits across the draft tokens. Second, to prevent the collection of these logits from becoming a new bottleneck, we introduce a "Collect-2-Query" mechanism. Instead of dumping the logits for every token to the GPU memory, we extract them only from the first draft token and the bonus token. These boundary tokens have the maximum positional distance and can effectively capture the diversity across all draft tokens. This significantly reduces the memory bandwidth requirement while maintaining high drafting accuracy. Finally, to maximize end-to-end throughput, we employ a systematic three-step hyperparameter tuning strategy to optimally balance the sparsity ratio and the number of draft tokens.

We implement Vegas within vLLM (Kwon et al., 2023) and evaluate it using diverse LLMs (e.g., the Qwen3 family and gpt-oss-20b) with different context lengths. Vegas achieves a $1.25\times$–$2.81\times$ speedup in decoding throughput over default vLLM, and a $1.15\times$–$1.29\times$ speedup over state-of-the-art sparse attention-based self-speculative decoding methods, driven by superior drafting accuracy and minimal KV selection overhead. The performance gains of Vegas scale with context length, as our KV selection strategy can achieve high accuracy without a proportional increase in overhead. We summarize our key contributions as follows:

- We propose Vegas, a self-speculative decoding mechanism that leverages verification to guide drafting.

- We design a low-overhead KV selection algorithm to achieve high drafting accuracy with minimal cost.

- We implement Vegas in vLLM and demonstrate significant throughput improvements over state-of-the-art baselines on diverse benchmarks.

## 2. Background and Motivation

### 2.1. Self-Attention and KV Cache

**Attention mechanism.** Transformer-based large language models (LLMs) generate text auto-regressively, where each generated token depends on all preceding tokens. This dependency is modeled via the self-attention mechanism (Vaswani et al., 2017). Specifically, the full context (preceding tokens and the current token) is projected into Key ($K \in \mathbb{R}^{n \times d}$) and Value ($V \in \mathbb{R}^{n \times d}$) matrices, while the current token is projected into a Query vector ($q \in \mathbb{R}^d$). Here, $n$ denotes the total token count (i.e., context length) and $d$ represents the hidden dimension of the attention heads. The model then computes attention weights[1] by querying the key vector of each token, and uses these weights to aggregate the value vectors into the attention output:

$$\text{Attention}(q, K, V) = \text{softmax}\left(\frac{qK^\top}{\sqrt{d}}\right) V \qquad (1)$$

**KV cache.** To avoid recomputing the keys and values of all preceding tokens when generating each new token, *KV cache* is employed to memoize these states in GPU memory (Kwon et al., 2023). Thus, LLM inference typically involves two phases: a *prefill* phase processes the input tokens in parallel to produce the initial KV cache of all input tokens, and a *decoding* phase generates subsequent tokens one by one while utilizing and updating the KV cache.

### 2.2. Sparse Attention for LLM Decoding

**KV cache access bottleneck.** The context length of LLMs grows rapidly from 4K to 128K (Llama Team, 2024), 1M (Yang et al., 2025b), and beyond (Gemini Team, 2024). As the KV cache size grows linearly with context length, the large KV cache incurs significant overhead (Tang et al., 2024; Sadhukhan et al., 2025). For example, serving Qwen3-8B (Yang et al., 2025a) on an H100 (batch size 4, 128K context) requires 36.9 ms per decoding step (see §5.1 for detailed experimental setup). This overhead is mainly due

---

[1]In this paper, we refer to $qK^\top$ as attention *logits* and softmax($qK^\top/\sqrt{d}$) as attention *weights*.

to KV cache access: the 72 GB KV cache dictates a theoretical minimum data transfer time of 18.5 ms (>50% of total decoding step time) (NVIDIA, 2024).

**Sparse attention.** To mitigate the KV cache access bottleneck in LLM decoding, sparse attention techniques have been proposed (Xiao et al., 2024b; 2025; Li et al., 2024; Tang et al., 2024). These methods are built on the insight that, during decoding, typically only a subset (e.g., 5%) of KV cache entries are critical for producing accurate outputs (Tang et al., 2024; Chen et al., 2026). Depending on how they identify critical KV entries, these methods can be categorized as *query-agnostic* or *query-aware* sparsity.

Query-agnostic sparsity selects KV entries using a fixed pattern without considering the current token, such as retaining recent tokens (sliding window (Beltagy et al., 2020)) or tokens that have historically accumulated high attention weights (e.g., StreamingLLM (Xiao et al., 2024b), H2O (Zhang et al., 2023)). This imposes minimal computational overhead. However, since the critical KV entries can vary across decoding steps (Chen et al., 2026), fixed patterns often fail to retrieve tokens that were previously unimportant but have since become relevant.

Query-aware sparsity (e.g., Quest (Tang et al., 2024)) addresses this limitation by estimating the relevance of all previous tokens to the current token at every decoding step. Such methods can retrieve sparse but critical information from long contexts. However, these methods incur higher computational overhead, as KV criticality estimation must be performed for every generated token.

Once the critical KV entries are selected, attention weights are computed only for the selected keys, and the attention output is a weighted average of the corresponding values. This significantly reduces KV cache access. However, both query-agnostic and query-aware sparsity methods inherently risk degrading output quality since they discard part of the context (Li et al., 2025a; Zhang et al., 2025b).

### 2.3. Self-Speculative Decoding with Sparse Attention

To mitigate the KV cache access bottleneck while ensuring output quality, self-speculative decoding with sparse attention has been proposed as a lossless acceleration technique (Sadhukhan et al., 2025; Sun et al., 2024). As shown in Figure 1a, the core idea is to leverage the original model with sparse attention to draft $\gamma$ subsequent tokens sequentially. These draft tokens[2] are then verified in a single for-

---

[2]In Speculative Sampling (Leviathan et al., 2023), the output token from the final drafting step is also decoded during verification, resulting in $\gamma + 1$ tokens being computed. This additional token is used when all preceding $\gamma$ draft tokens are accepted, and is thus referred to as the *bonus* token. In the remainder of this paper, we regard the bonus token as a draft token by default. A detailed illustration of speculative decoding is provided in Appendix A.

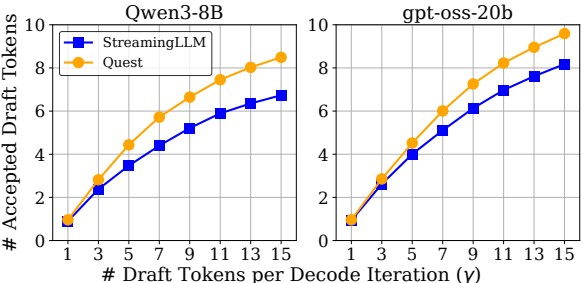

*Figure 2.* The average number of accepted draft tokens (excluding the bonus token) per decode iteration using StreamingLLM and Quest as drafters in self-speculative decoding. Quest achieves higher drafting accuracy via its query-aware attention sparsity compared to the query-agnostic StreamingLLM. Results are profiled on LongBench-v2 with 7% of KV cache entries selected.

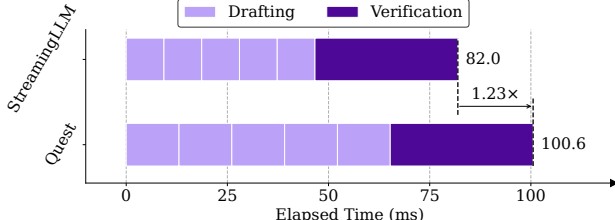

*Figure 3.* Execution time breakdown of one decoding iteration (drafting 5 tokens + verification) using StreamingLLM and Quest as drafters. Results are profiled on LongBench-v2 when running Qwen3-8B with a batch size of 4 on an H100 GPU.

ward pass with full attention, which only requires loading the full KV cache *once*. Draft tokens that align with the full attention output are accepted, while the rest are discarded. The process then repeats just as standard auto-regressive decoding, with the next drafting phase resuming from the first discarded token (in the rest of this paper, we refer to this token as the *rejected* token to distinguish it from other *discarded* tokens). This approach ensures that the final output has the same distribution as vanilla full-attention decoding.

The performance benefits of these self-speculative decoding methods are greatly influenced by the sparsity algorithm employed, in two dimensions: drafting accuracy (i.e., the percentage of draft tokens accepted) and KV selection overhead. To understand this relationship, we study the state-of-the-art method MagicDec (Sadhukhan et al., 2025) with two representative sparse attention algorithms: StreamingLLM (query-agnostic) and Quest (query-aware). As shown in Figure 2, Quest consistently achieves higher drafting accuracy on both Qwen3-8B and gpt-oss-20b (OpenAI, 2025b) across various $\gamma$, achieving a 22% improvement in the average number of tokens decoded per iteration (including the bonus token) when $\gamma = 5$. However, as shown in Figure 3, KV selection inflates the iteration duration by 23%. As a result, the computational cost outweighs the drafting gains, leading to a 1.7% reduction in total decoding throughput relative to StreamingLLM (204.5 vs. 208.0 tokens/s).

Fundamentally, there is a trade-off between drafting accuracy and KV selection overhead: achieving higher drafting accuracy requires more precise KV selection, which typically incurs higher computational cost.

## 3. Vegas Design

### 3.1. Core Idea

To tackle the trade-off between drafting accuracy and KV selection overhead, our core insight is to co-design drafting and verification. In existing frameworks, verification merely provides a binary signal (i.e., accept or reject) without offering guidance on *how to improve* drafting. Even though the verification step performs full attention and inherently calculates the exact criticality (i.e., the attention weights/logits) of every KV entry, current methods overlook this "free oracle" as they treat the drafting stage as a standalone module.

Inspired by these observations, we propose Vegas, a novel self-speculative decoding method with ***ve**rification-**g**uided **s**parse **a**ttention*. As shown by Figure 1b, the verification phase serves two purposes: verifying draft tokens and identifying critical KV entries as a byproduct of the full attention computation. In the subsequent drafting phase, sparse attention is computed over these selected entries.

We present Vegas in the remainder of this section as follows. §3.2 describes how Vegas selects KV entries based on intermediate results derived from verification. §3.3 introduces algorithmic optimizations designed to minimize the overhead of deriving these intermediate results. Finally, §3.4 discusses how to tune the key hyperparameters for achieving the best end-to-end performance.

### 3.2. Verification-Guided KV Selection

Leveraging verification to select KV entries that lead to high drafting accuracy is non-trivial: We need to predict KV entries critical to *future* draft tokens based on the attention results computed by the verification of *current* draft tokens.

An intuitive approach is to select the KV entries with the highest attention weights for the last accepted token, since these entries are likely to remain critical in the next few decoding iterations (Wu et al., 2025). However, we observed that, although this strategy achieves high acceptance probabilities[3] for the first few tokens drafted in each drafting phase, the acceptance probability drops significantly for the later tokens in the draft token chain, as shown by Figure 4. This is because selecting KV entries based on a single token risks overfitting to that token's specific attention pattern, thereby failing to capture the general local context needed

---

[3]The acceptance *probability* of a draft token is defined as the probability of the token being accepted given that all previous draft tokens are accepted (Leviathan et al., 2023).

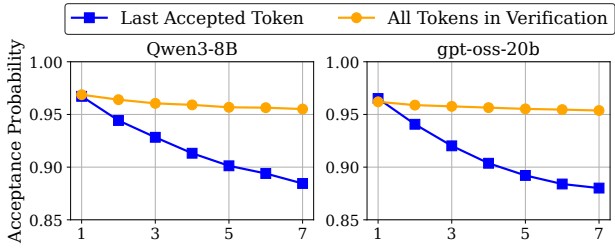

*Figure 4.* Comparison of two KV selection strategies. Selecting KV entries with the highest attention weights for the last accepted token results in a rapid decay in acceptance probability as the draft token's position in the draft chain increases. In contrast, selecting KV entries that maximize the overall attention weight coverage across all draft tokens maintains consistently high acceptance probabilities. The y-axis starts at 0.85 for clarity. Results are profiled on LongBench-v2 with 7% of KV entries selected.

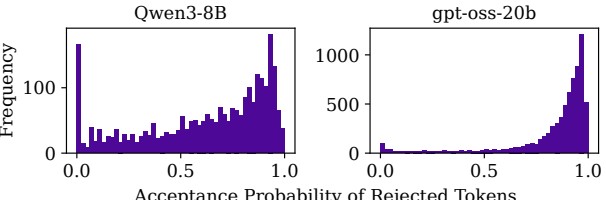

*Figure 5.* Distribution of acceptance probability of rejected tokens. Many rejected tokens are sampled from a distribution that closely matches the one predicted by full attention.

for subsequent tokens in the draft chain.[4]

Instead of selecting KV entries based on the last accepted token, maximizing the coverage of attention weights across *all* draft tokens (including both accepted and discarded ones) yields much higher acceptance probabilities (see Figure 4). While this approach may slightly reduce the acceptance probabilities of the first draft token compared to the last-accepted-token method (which aligns with our overfitting hypothesis), it avoids the rapid decay of acceptance probability for subsequent draft tokens.

Furthermore, including the discarded tokens improves drafting accuracy. This is because many draft tokens possessing a high acceptance probability are rejected due to probabilistic sampling (Leviathan et al., 2023), as shown in Figure 5. Consequently, these rejected tokens often remain semantically aligned with the target generation. For instance, a draft token "on" might be corrected to "upon" in the sequence "the cat sat [on/upon] the mat". Conditioned on such a semantically similar token, the remaining discarded tokens (e.g., "the mat") are likely to match the tokens generated using full attention. Thus, their attention weights also help with the KV criticality estimation. Our ablation experiment

---

[4]Throughout this paper, we use "overfit" to indicate that the selected KV entries are overly biased toward the attention pattern of a single query token, analogous to its use when a model overfits its training data and fails to generalize to unseen examples.

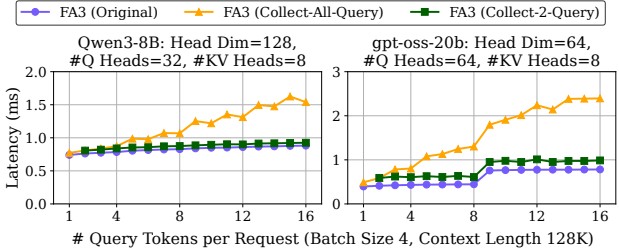

Figure 6. Overhead of attention logit collection in FlashAttention-3 (FA3). Collecting attention logits from all query tokens incurs significant overhead due to memory-bandwidth contention, which scales with the number of query tokens. In contrast, our "Collect-2-Query" approach reduces this overhead to a minimal, constant level by collecting logits from only the first draft and bonus tokens.

shows that ignoring these discarded tokens decreases the average accepted tokens per decoding iteration by 3%–14% when $\gamma = 7$ for both Qwen3-8B and gpt-oss-20b.

We formalize the KV entry selection strategy of "maximizing the coverage of attention weights across all draft tokens" as follows. Let $p$ denote the length of the token prefix on which the draft tokens are conditioned. During the verification phase, an attention logit[5] matrix $L^{(h)} \in \mathbb{R}^{(\gamma+1) \times p}$ is computed for each query head $h$. A subset of $k$ prefix tokens that maximizes the cumulative attention logit across all draft tokens is selected:

$$T = \underset{\substack{T \subseteq P \\ |T|=k}}{\arg\max} \sum_{t=1}^{\gamma+1} \sum_{h \in H} \sum_{i \in T} l_{ti}^{(h)} \tag{2}$$

where $T$ is the set of indices for the selected prefix tokens, $P = \{1, 2, \dots, p\}$ is the set of all prefix token indices, $H$ denotes the set of all query heads, and $l_{ti}^{(h)}$ represents the entry at row $t$ and column $i$ of $L^{(h)}$. In the subsequent drafting phase, prefix tokens are sparsified based on $T$ while the remaining tokens are kept.

### 3.3. Low-Overhead Attention Logit Collection

Equation 2 requires collecting the attention logits of all draft tokens. In practice, this incurs significant execution time overhead in the FlashAttention-3 (FA3) kernel (Shah et al., 2024) (a highly-optimized attention kernel employed by most LLM inference engines today). As shown in Figure 6, the latency of the original FA3 kernel grows only marginally as we increase the number of query tokens[6] (equal to $\gamma + 1$ for the drafting phase), since the kernel is bottlenecked by loading the KV cache, whose size is independent of query

---

[5]While attention weights are commonly used for selecting critical KV entries (Zhang et al., 2023; Li et al., 2024; Oren et al., 2024; Lin et al., 2025), we use attention logits since they are computationally cheaper to collect and provide similar drafting accuracy in practice (see Appendix B).

[6]Except for a distinct latency jump from 8 to 9 tokens for gpt-oss-20b, this is due to kernel tiling boundaries.

---

Table 1. The average number of draft tokens accepted per iteration obtained using the two proposed KV selection methods. Results are profiled on LongBench-v2 with 7% of KV entries selected.

| Method | Qwen3-8B | | gpt-oss-20b | |
|---|---|---|---|---|
| | $\gamma = 7$ | $\gamma = 11$ | $\gamma = 7$ | $\gamma = 11$ |
| All Draft Tokens | 6.13 | 8.91 | 6.25 | 9.27 |
| 1st Draft + Bonus | 6.11 | 8.72 | 6.24 | 9.22 |

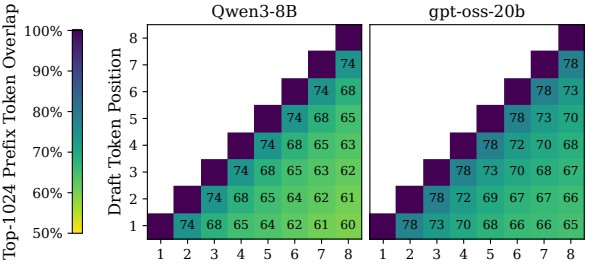

Figure 7. The pairwise overlap ratio of the top-1024 prefix tokens selected by each draft token ($\gamma = 7$). The ratio diminishes as the positional gap between draft tokens increases. Results are averaged across 4 samples from LongBench-v2 and all attention blocks.

tokens. In contrast, the overhead of collecting attention logits for all query tokens scales significantly with the number of query tokens, resulting in 53% overhead for Qwen3-8B and 189% for gpt-oss-20b when using 12 query tokens.

To mitigate this overhead, we find that reducing the number of draft tokens for collecting attention logits will not hurt the drafting accuracy significantly. Remarkably, we observe that using only *the first draft token and the bonus token* leads to a drafting accuracy comparable to using all draft tokens (see Table 1). Since we only collect the attention logits of two tokens, the overhead can be reduced to as low as 5% for Qwen3-8B and 37% for gpt-oss-20b (see Figure 6). Furthermore, our "Collect-2-Query" approach is much more scalable: the overhead does not increase as we increase $\gamma$.

The rationale for selecting the first draft token and the bonus token lies in the observation that KV entries selected by consecutive tokens exhibit high overlap. As shown in Figure 7, this overlap diminishes as the distance between tokens increases. Therefore, to avoid overfitting to a consecutive range of tokens, it is preferable to select the pair of tokens with the maximum positional distance. In our design, these correspond to the first draft token and the bonus token. We formalize Vegas's KV selection strategy as follows:

$$T = \underset{\substack{T \subseteq P \\ |T|=k}}{\arg\max} \sum_{h \in H} \sum_{i \in T} (l_{1i}^{(h)} + l_{(\gamma+1)i}^{(h)}) \tag{3}$$

In practice, $T$ is computed by first averaging the attention logits of the first draft token and the bonus token. We then average this result across all query heads ($h$) to obtain a criticality score for each prefix token. Finally, $T$ is obtained by selecting the $k$ tokens with the highest scores.

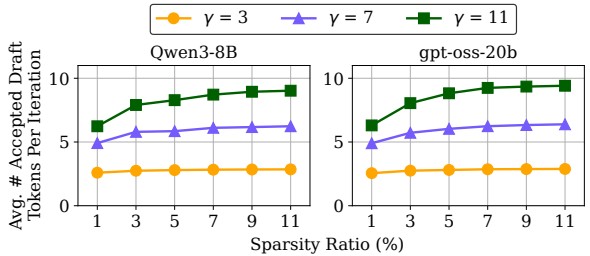

*Figure 8.* Sensitivity of the number of draft tokens accepted per decode iteration to the sparsity ratio. As the sparsity ratio increases, the average number of accepted tokens initially increases and then stabilizes. Results are profiled on LongBench-v2.

### 3.4. Hyperparameter Tuning

The performance of self-speculative decoding with sparse attention depends on two critical hyperparameters: the number of tokens drafted per decode iteration $\gamma$ and the sparsity ratio[7] (the number of KV entries selected for sparse attention over the total number of KV entries).

**Sparsity ratio.** Figure 8 shows the number of accepted draft tokens per decoding iteration as we vary the sparsity ratio. As the sparsity ratio increases, the average number of accepted draft tokens rises initially and then reaches a relatively stable level. This occurs because once the selected KV subset covers the majority of critical KV entries, adding further non-critical entries yields diminishing returns.

**Number of draft tokens ($\gamma$).** Intuitively, $\gamma$ should align with the expected number of accepted tokens per decoding iteration, which depends on the sparsity ratio. If $\gamma$ far exceeds this expected count, we waste computation generating tokens that are likely to be rejected. Conversely, if $\gamma$ is lower than the expected count, we fail to fully exploit the potential of correct speculations.

**Vegas hyperparameter tuning strategy.** Based on the above observations, we adopt a three-step tuning strategy. (1) Given a model and a sample set of requests, we sweep the sparsity ratio to identify the point where the number of accepted tokens stabilizes. This sweep can be performed with a sufficiently large $\gamma$. (2) With the selected sparsity ratio, we sweep $\gamma$ to maximize decoding throughput. (3) With $\gamma$ fixed, we can further adjust the sparsity ratio to balance the trade-off between drafting latency and accuracy.

## 4. Implementation

**Serving framework integration.** We implement Vegas within vLLM (Kwon et al., 2023). We intercept calls to vLLM's `flash_attn_varlen_func` using a custom hook, which dynamically dispatches the correct attention kernel: for drafting, it employs sparse attention; for verifica-

tion, it uses our instrumented attention kernel for collecting attention logits. To implement the drafting stage, we create a custom Python class to implement vLLM's speculative decoding proposer API, which calls the forward pass of the target model (with sparse attention) multiple times and returns the draft tokens. To enable sparse attention, we repurpose the PagedAttention kernel (`flash_attn_varlen_func`) by using a page size of 1 token and passing the selected token indices as page indices, such that it will only perform computation on the selected tokens. This token-level selection maximizes drafting accuracy and incurs no performance overhead in the attention kernel: the kernel latency is nearly identical from page size 1 to 16 (see Appendix D).

**Attention kernel implementation.** We instrument vLLM's FlashAttention-3 (Shah et al., 2024) kernel to collect attention logits. We modify the kernel to write the attention logits into the global memory (i.e., HBM) after causal or local masking and before softmax normalization. Logits are cast to BF16 to minimize HBM bandwidth overhead and written directly to HBM without occupying the precious on-chip shared memory. When logit collection is disabled, the kernel retains its original behavior with negligible overhead from the added conditional checks. This ensures modularity and facilitates integration into the vLLM ecosystem.

## 5. Evaluation

### 5.1. Experimental Setup

**Models.** We evaluate open-source LLMs with diverse model architectures and sizes, including Qwen3-4B, Qwen3-8B, Qwen3-30B-A3B-Thinking-2507-FP8 (hereinafter Qwen3-30B) (Yang et al., 2025a), and gpt-oss-20b (OpenAI, 2025b). For gpt-oss-20b, we apply MXFP4 quantization for the mixture-of-experts weights. Since its attention blocks alternate between banded window and fully dense patterns, we apply sparse attention only to dense attention blocks. For each LLM, we follow the sampling parameters recommended by its HuggingFace model card (see Appendix E.1).

**Hardware.** All experiments are conducted on a server equipped with two NVIDIA H100 NVL GPUs (94 GB).

**Baselines.** To ensure a fair comparison, all baselines are implemented within vLLM: (1) *vLLM*: The standard serving framework with its default configuration, serving as our baseline for vanilla decoding. (2) *MagicDec* (Sadhukhan et al., 2025): A framework utilizing self-speculative decoding with sparse attention. We evaluate it using two representative KV selection methods: StreamingLLM (Xiao et al., 2024b) (*MagicDec-Stream*) and Quest (Tang et al., 2024) (*MagicDec-Quest*). (3) *SpecExtend* (Cha et al., 2025): An algorithm leveraging the last accepted token's attention weights from the target LLM's final attention block to guide the sparsity of an auxiliary drafter. As it is origi-

---

[7]We use the sparsity ratio rather than fixed token budgets because it adapts to varying sequence lengths (see Appendix C).

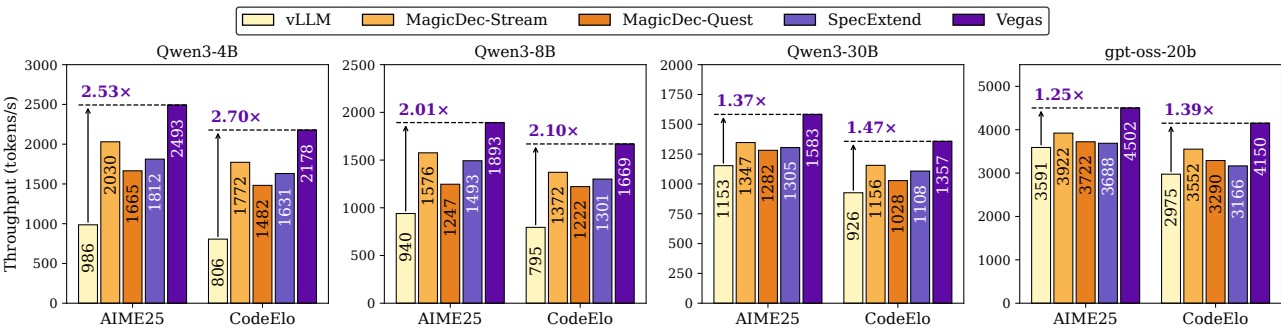

*Figure 9.* Decoding throughput with datasets AIME25 and CodeElo (long reasoning outputs with short input questions).

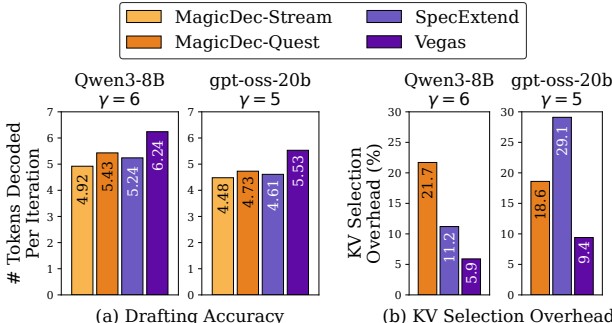

*Figure 10.* Drafting accuracy and KV selection overhead of Vegas compared to other methods on AIME25 dataset. We show Qwen3-8B and gpt-oss-20b as examples for studying the trends. In (a), the corrected / bonus token is included. In (b), the overhead is with respect to the ideal iteration latency (i.e., draft $\gamma$ tokens and verify, without selecting KV cache entries). We do not include MagicDec-Stream in (b) because sliding window attention incurs negligible KV selection overhead.

nally designed for a separate draft model, to ensure a fair comparison under self-speculative decoding, we apply its guidance mechanism layer-wise at every attention block. We set $\gamma$ and sparsity ratio for each design following §3.4 (see Appendix E.2 for the hyperparameter configuration).

## 5.2. Reasoning Workloads with Short Input Context

We evaluate Vegas on AIME25 (Zhang & Math-AI, 2025) and CodeElo (Quan et al., 2025), spanning both math and coding domains. These benchmarks are characterized by concise input contexts (e.g., 182.0 tokens on average for AIME25) and extensive chain-of-thought (Wei et al., 2022) reasoning outputs (averaging 19,680 tokens[8]). Datasets are replayed until a stable decoding throughput is measured (Appendix E.3). All the experiments are run on a single H100 GPU, with the maximum batch size set to 128, which we found is enough to saturate the available KV cache capacity.

**Overall decoding throughput.** Figure 9 shows that Vegas achieves a 1.25×–2.70× speedup in decoding throughput over vLLM. MagicDec-Stream remains the most competitive baseline, largely due to its lightweight query-agnostic KV selection strategy. MagicDec-Quest performs consis-

tently worse than MagicDec-Stream, as its query-aware KV selection incurs high overhead that outweighs the benefits from its higher drafting accuracy. SpecExtend achieves a lower drafting accuracy compared to MagicDec-Quest, as it decides KV pages based on the last accepted token before drafting and does not update them. Vegas outperforms MagicDec-Stream by 1.15×–1.23×, as it achieves higher drafting accuracy with comparable KV selection overhead.

For gpt-oss-20b, all self-speculative decoding methods yield relatively smaller speedups over vLLM. This stems from the model's architecture, where only half of the transformer blocks employ full attention. Consequently, applying sparse attention only reduces the per-drafting step latency to 48% of the vanilla decoding latency.

**Drafting accuracy.** We analyze Vegas's drafting accuracy in Figure 10a. We fix $\gamma$ to the optimal value for Vegas to ensure a consistent experimental setup. Vegas achieves higher drafting accuracy compared to MagicDec-Stream and MagicDec-Quest, primarily due to its token-granular KV selection. Vegas also outperforms SpecExtend in accuracy, as our multi-token selection strategy prevents the overfitting issues observed when relying on a single token (see §3.2).

**KV selection overhead.** Figure 10b shows the KV selection overhead. For Qwen3-8B, MagicDec-Quest incurs 21.7% overhead due to re-estimating KV criticality in every drafting step. SpecExtend only needs to select critical KV entries once per decoding iteration, which leads to a lower overhead (11.2%). However, this one-time overhead is amplified to 29.1% by gpt-oss-20b, whose attention layers feature a higher query-to-KV head ratio (more logits to collect) and smaller head dimensions (relatively more metadata per KV entry). SpecExtend incurs higher overhead than Vegas since it conditions its KV selection on the last accepted token, which is indeterminate during the attention kernel execution. Hence, it is forced to collect attention logits for all draft tokens and maintain them in HBM until the last accepted token is determined. Vegas incurs only 5.9%–9.4% overhead, which is much smaller than all other query-aware methods. This overhead is acceptable given the overall speedup brought by the drafting accuracy improvement.

---

[8]Generated by Qwen3-8B and measured with its tokenizer.

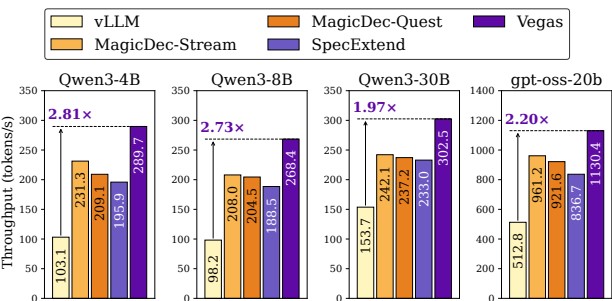

*Figure 11.* Decoding throughput with LongBench-v2 (long input context and reasoning outputs).

## 5.3. Reasoning Workloads with Long Input Context

We further evaluate Vegas using LongBench-v2 (Bai et al., 2025), a benchmark designed to assess model capabilities over extensive contexts. We sample requests with input lengths ranging from 96K to 120K tokens (measured by each model's respective tokenizer) for evaluation. To accurately measure decoding throughput without prefill interference, we employ a prefill-decoding disaggregation strategy (Zhong et al., 2024) across two GPUs. The decoding GPU operates with maximum batch sizes tailored to each model's capacity: 4, 4, 5, and 20. RoPE (Su et al., 2024) scaling is applied to Qwen3-4B/8B with a factor of 4.

On such long-context tasks, the performance advantage of Vegas becomes more apparent compared to baselines. StreamingLLM inherently struggles due to low drafting accuracy as it discards historical context. Quest becomes more competitive by retaining historical recall capabilities, but suffers from KV selection overhead. SpecExtend fails to benefit from generating longer draft sequences due to the decline in drafting accuracy for later tokens. In contrast, Vegas maintains high acceptance rates even with extended contexts, achieving 18%–29% higher decoding throughput than the most competitive baselines.

## 6. Related Work

**Sparse attention.** Many sparse attention algorithms have been proposed to reduce the KV cache size and accelerate attention computation. Query-agnostic sparsity methods maintain fixed attention patterns (Xiao et al., 2024b; 2025) or KV selection decisions (Li et al., 2024; Yang et al., 2024) in a manner that is agnostic to the current query token. Query-aware sparsity methods select KV entries based on their estimated importance to the current token (Tang et al., 2024; Xiao et al., 2024a; Liu et al., 2025a; Zhang et al., 2025a; Zhu et al., 2025; Wu et al., 2025; Chen et al., 2025b). These methods improve LLM serving performance at the cost of degraded output quality. The training-native sparsity approach, such as NSA (Yuan et al., 2025), integrates sparse attention directly into the model pretraining process to achieve improved model accuracy. However, this ap-

proach is model-specific and has limited applicability to other off-the-shelf models. Our solution, by contrast, is a training-free method that couples speculative decoding with co-designed sparse attention, thereby preserving generation quality while achieving remarkable speedup.

**Speculative decoding with auxiliary draft models.** Prior works have proposed using a dedicated small model to generate draft tokens for speculative decoding (Leviathan et al., 2023; Xia et al., 2023; Chen et al., 2023). The draft model can be an n-gram-based token predictor (Saxena, 2023; Fu et al., 2024), another smaller LLM, or an LLM distilled from the target model (Zhou et al., 2024). These methods typically suffer from a low acceptance rate due to the limited output quality of the small model. To mitigate this issue, several works proposed training a dedicated draft model to better align with the specific target model (Liu et al., 2024; Yang et al., 2025a; Li et al., 2025b). Our solution leverages self-speculative decoding, achieving a high acceptance rate without training or deploying an auxiliary model.

**Self-speculative decoding.** Prior works proposed using the target model with sparse attention techniques (Sadhukhan et al., 2025; Sun et al., 2024; Chen et al., 2025a; Ji et al., 2025) as the draft model in speculative decoding. None of them uses the attention results from the verification stage to improve drafting accuracy at every decoding iteration. SpecExtend (Cha et al., 2025) uses the attention weights in the last layer of the target model to guide the draft model's KV selection. It suffers from suboptimal drafting accuracy because it selects KV entries based solely on the last accepted token. Several works propose selectively skipping layers during drafting (Elhoushi et al., 2024; Zhang et al., 2024; Xia et al., 2025) or using a quantized version of the target model as the drafter (Tiwari et al., 2025). They can be applied in conjunction with our sparse attention mechanism to further speed up LLM inference. Most recently, SparseSpec (Zhao et al., 2025) proposed a similar verification-guided approach. However, it selects critical KV entries based on the attention weights of all draft tokens, which introduces nontrivial rematerialization overhead. In contrast, we identified the opportunity to use the attention logits from only the first draft token and the bonus token for KV selection, substantially reducing this computational cost.

## 7. Conclusion

We presented Vegas, a self-speculative decoding approach that co-designs drafting and verification to accelerate long-context LLM inference. Unlike prior work that treats drafting as a standalone process, Vegas identifies critical KV entries as a byproduct of verification and uses only these entries for drafting. Vegas is training-free and accelerates decoding losslessly. It achieves a 1.15×–1.29× speedup in decoding throughput over state-of-the-art baselines.

## Acknowledgements

We thank the anonymous reviewers for their insightful comments and feedback. We also thank Denis Filimonov and Giannis Karamanolakis at Amazon for valuable discussions, Ziyuan Lin for implementing Quest in our testbed, and Steven Lumetta, Jiankun Wang, and Jiahuan Yu for helpful discussions on implementing attention logit collection in FlashAttention. This work was partially supported by AICE Center (Amazon-Illinois Center on AI for Interactive Conversational Experiences) and NSF CAREER CNS-2144796.

## Impact Statement

This paper introduces Vegas, a self-speculative decoding framework that significantly accelerates long-context LLM inference. By co-designing the drafting and verification phases, Vegas mitigates the memory bottleneck inherent in processing massive KV caches, achieving significant throughput improvement while maintaining the lossless generation quality of the original model.

**Computational efficiency and environmental impacts.** A positive impact of this work is the reduction of computational resources required for LLM serving. As the context length of LLMs scales to millions of tokens, the energy cost of inference becomes a critical concern. The performance benefits of Vegas directly translate to energy savings. Furthermore, Vegas can be immediately deployed on existing LLM models without retraining or fine-tuning specialized draft models, which typically incurs substantial energy consumption and associated carbon footprint.

**Reliability and safety.** A common risk of sparse attention techniques is degraded model output quality (e.g., increased hallucinations due to missing context). Vegas guarantees no degradation in output quality while exploiting the performance benefits of sparse attention. This reliability is crucial for high-stakes applications (e.g., legal or medical analysis) where efficiency cannot come at the cost of accuracy.

**Risks and dual-use implications.** While Vegas improves efficiency, it does not alter the fundamental capabilities of the underlying LLMs. The ethical implications of deploying Vegas are identical to those of the base model itself; our method introduces no new safety risks or biases. As increased inference efficiency acts as a multiplier for both beneficial and harmful applications, we encourage users to deploy Vegas alongside the same safety mechanisms that would be employed for the underlying models.

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

# A. Background on Speculative Decoding

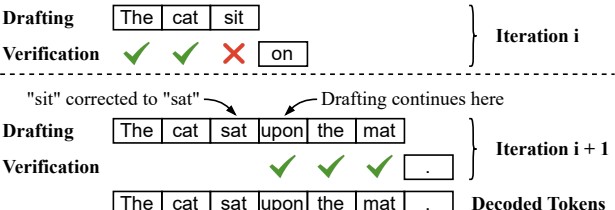

*Figure 12.* An example of speculative decoding over two decoding iterations, with $\gamma = 3$. A check mark denotes accepting a token, and a cross denotes rejecting a token.

We use Figure 12 to provide a clearer definition of accepted, rejected, bonus, and corrected tokens. In the $i$-th decoding iteration, three tokens ("The", "cat", and "sit") are drafted. During verification, "The" and "cat" are accepted, but "sit" is rejected. Therefore, the bonus token "on", which was decoded conditioned on "sit", is discarded. Speculative Sampling corrects the rejected token "sit" to "sat", and drafting in the next iteration starts right after "sat". We refer to "sat" as the *corrected* token. In the $(i{+}1)$-th iteration, three more tokens are drafted, and a bonus token "." is decoded during verification. Since all three draft tokens are accepted, the bonus token "." is also accepted, and the next drafting phase starts after it.

For the "last accepted token" that is used in our discussion, its definition depends on the verification outcome: it is the corrected token if any draft token is rejected (e.g., "sat" in the $i$-th iteration), or the bonus token if all draft tokens are accepted (e.g., "." in the $(i{+}1)$-th iteration).

# B. Attention Weights vs. Attention Logits

As discussed in §3.2, Vegas uses attention logits ($qK^{\top}$) as the metric to select critical KV entries, which incurs low overhead and achieves high drafting accuracy.

Alternatively, attention weights, $\mathrm{softmax}(qK^{\top}/\sqrt{d})$, can be used as the metric for KV selection. However, collecting attention weights incurs higher computational overhead. State-of-the-art attention kernels (Shah et al., 2024; Ye et al., 2025) employ online softmax, and the final attention weights cannot be directly extracted from the intermediate computation. Therefore, collecting attention weights requires an additional rematerialization step based on the attention logits. In our code release, we provide both logit-based and weight-based implementations. For the latter, we introduce a custom kernel that fuses attention weight rematerialization with logit/weight averaging to minimize its overhead.

In our experiments, the choice between attention weights and logits does not significantly affect drafting accuracy. Because softmax is a monotonic function, both metrics yield identical token rankings within any single query head. These

*Table 2.* FlashAttention-3 kernel latency ($\mu$s) for sparse attention at various page sizes, profiled on an H100 GPU (batch size 16).

| Model | Page Size (token(s) per page) | | | | |
|---|---|---|---|---|---|
| | 1 | 2 | 4 | 8 | 16 |
| Qwen3-8B | 197.6 | 197.6 | 197.1 | 198.1 | 198.3 |
| gpt-oss-20b | 131.0 | 130.1 | 130.1 | 130.3 | 130.8 |

rankings only diverge when the values are averaged across multiple query heads. Yet this divergence is insignificant when selecting a small subset of critical tokens for two primary reasons. First, the variance of attention logits is relatively consistent across query heads, preventing any single outlier head from dominating the averaged ranking. Second, recent research (Agarwal et al., 2024) demonstrates a strong correlation among the attention patterns of different query heads (i.e., multiple query heads frequently assign high attention logits, and thus weights, to the same prefix tokens). Because these tokens are deemed critical across several redundant heads, they are reliably selected, no matter the cross-head average is computed using logits or weights.

# C. Sparsity Ratio vs. Fixed Token Budget

While a fixed token budget for sparse attention is commonly used by some prior works (Tang et al., 2024; Sadhukhan et al., 2025), we instead parameterize sparsity as a ratio (the number of selected KV entries over the total) because it adapts to varying sequence lengths. Recent work (Song et al., 2025) provides theoretical evidence that maintaining accuracy requires selecting more tokens as context length grows. We also observe this empirically: for Qwen3-8B on LongBench-v2, a 3% sparsity ratio (around 3000 tokens) is not enough to reach the point where Vegas's drafting accuracy stabilizes (Figure 8), whereas a 3000-token budget is more than sufficient for a request from AIME25. A fixed token budget thus cannot serve both regimes well, whereas a sparsity ratio automatically scales the token budget with each request's context length.

# D. Page Size for Sparse Attention

As described in §4, Vegas enables sparse attention by re-purposing the PagedAttention kernel with a page size of 1 token, passing the selected token indices as page indices. A page size of 1 provides token-level selection granularity, maximizing drafting accuracy. Although finer granularity could result in poor data locality and longer attention kernel latency, we show that this is not the case.

We benchmark the FlashAttention-3 kernel with various page sizes on an H100 GPU. We set the batch size to 16 (to amortize kernel launch overhead), allocate a KV cache buffer for 16×128K tokens, and select 16×8192 tokens

(each request has 8192 / page_size pages) with page positions randomly chosen from the full buffer. Table 2 reports the kernel latency. The latency is almost identical across page sizes from 1 to 16 (vLLM's default). The differences are negligible mainly because Vegas selects the same tokens for all KV heads. Under the NHD KV layout (tensor shape [num_pages, page_size, num_kv_heads, head_dim]), each token's KV cache forms a contiguous buffer of 4 KB for Qwen3-8B and 2 KB for gpt-oss-20b, which already provides sufficient memory locality.

## E. Detailed Experimental Setup

### E.1. Sampling Parameters

For all experiments in §5, we follow each model's recommended sampling parameters. For Qwen3 models, we set `Temperature` to 0.6, `TopP` to 0.95, `TopK` to 20, and `MinP` to 0. For gpt-oss-20b, we set `Temperature` to 1.0 and `TopP` to 1.0, with its default reasoning effort (medium).

### E.2. Self-Speculative Decoding Hyperparameters

*Figure 13.* Decoding throughput (token/s) of Vegas with various $\gamma$ and sparsity ratio. Profiled using Qwen3-8B on LongBench-v2.

For all designs and all models in our experiments in §5, we use our hyperparameter tuning strategy discussed in §3.4. Figure 13 showcases the decoding throughput as we sweep different values of $\gamma$ and sparsity ratio for Qwen3-8B on LongBench-v2. In general, once the sparsity ratio reaches a certain threshold (e.g., 3%), the choice of $\gamma$ impacts decoding throughput more significantly than the sparsity ratio itself. This is because the number of accepted tokens stabilizes after reaching this low level of sparsity (see Figure 8); as long as the sparsity ratio is low enough (e.g., $< 10\%$), altering this parameter will not cause a significant difference in the decoding iteration latency.

For the experiments in §5, our hyperparameter sweeping finds that a sparsity ratio of 7% achieves a near-optimal trade-off between drafting accuracy and overhead. Table 3 shows the selected $\gamma$ values. Vegas typically uses a larger $\gamma$ than other designs, as it has higher drafting accuracy and lower drafting latency (due to lower KV selection overhead), making a larger $\gamma$ more beneficial.

*Table 3.* The $\gamma$ values used in our experiments.

| Model & Design | Dataset | | |
|---|---|---|---|
| | AIME25 | CodeElo | LongBench-v2 |
| ***Qwen3-4B*** | | | |
| MagicDec-Stream | 4 | 4 | 5 |
| MagicDec-Quest | 4 | 4 | 5 |
| SpecExtend | 4 | 4 | 6 |
| Vegas | 6 | 6 | 9 |
| ***Qwen3-8B*** | | | |
| MagicDec-Stream | 4 | 4 | 5 |
| MagicDec-Quest | 4 | 4 | 5 |
| SpecExtend | 4 | 4 | 5 |
| Vegas | 6 | 6 | 9 |
| ***Qwen3-30B*** | | | |
| MagicDec-Stream | 3 | 4 | 3 |
| MagicDec-Quest | 3 | 3 | 3 |
| SpecExtend | 3 | 3 | 4 |
| Vegas | 5 | 5 | 7 |
| ***gpt-oss-20b*** | | | |
| MagicDec-Stream | 3 | 4 | 5 |
| MagicDec-Quest | 3 | 4 | 5 |
| SpecExtend | 3 | 3 | 4 |
| Vegas | 5 | 5 | 7 |

### E.3. Decoding Throughput Measurement

To measure a stable decoding throughput, we submit enough requests to the engine at once and wait until the serving engine reports a stabilized decoding throughput. Note that continuous batching (Yu et al., 2022) is enabled. For example, for AIME25, which has 30 distinct questions, we duplicate it 32 times and submit all requests at once. The decoding throughput stabilized after roughly 400 requests had been served, at which point we began measuring.

## F. Limitations

Self-speculative decoding with sparse attention only reduces KV cache accesses. Therefore, it only delivers significant benefits when the context is sufficiently long (Sadhukhan et al., 2025; Zhao et al., 2025). Thus, Vegas works the best for throughput-oriented long-context serving, where KV cache loading time is the primary bottleneck.

For latency-oriented scenarios (i.e., small batch sizes), Vegas may not provide enough benefits, but all sparse attention-based self-speculative decoding methods suffer similarly. However, as long as the attention computation outweighs that of the feed-forward layers, Vegas can deliver benefits. Empirically, for Qwen3-8B, Vegas starts to deliver benefits with a batch size of 8 when serving AIME25 and with a batch size of 1 for LongBench-v2.

