# OpenReview forum: "Vegas: Self-Speculative Decoding with Verification-Guided Sparse Attention"
_ICML.cc/2026/Conference — ICML 2026 regular_

### Official Review · Reviewer_V1N7 · 2026-03-07

**Soundness:** 2
**Presentation:** 2
**Significance:** 2
**Originality:** 2
**Overall Recommendation:** 2
**Confidence:** 4

**Summary:**

This paper proposes a self-speculative decoding framework with verification-guided sparse attention for accelerating long-context LLM inference. The method observes that the verification stage already computes full attention and therefore implicitly provides signals about which KV entries are important. Vegas leverages these to guide KV selection for the next drafting stage, iand it mproves drafting accuracy and reduce KV selection overhead. The authors implement Vegas in vLLM and demonstrate up to 2.81× throughput improvement over vanilla method in autoregressive decoding and moderate gains over existing sparse speculative decoding methods.

**Compliance With Llm Reviewing Policy:**

Affirmed.

**Key Questions For Authors:**

I have two questions and list below
1. How stable are the verification-derived KV importance signals across decoding steps?
The method assumes that KV entries identified during verification remain useful for subsequent draft tokens. Could the authors provide empirical analysis showing how stable these selected KV entries are across multiple decoding iterations?
If the signals are shown to be consistently predictive of future token attention patterns, it would strengthen the justification of the proposed approach.

2. How sensitive is Vegas to the sparse ratio and the number of draft tokens (γ)?
The paper proposes a tuning strategy for these parameters, but it is unclear how robust the method is to suboptimal configurations.
If Vegas performs well across a wide range of hyperparameters, this would increase confidence in its practical usability.

**Limitations:**

Partially. The authors discuss potential impacts and energy efficiency benefits in the Impact Statement. However, the discussion could be improved by considering broader implications of enabling more scalable LLM deployment (e.g., increased misuse potential due to reduced inference costs).

**Strengths And Weaknesses:**

For the good perspective, The paper addresses an important system bottleneck in long-context LLM inference, namely KV cache bandwidth and speculative decoding efficiency. and the key insight—reusing verification attention signals to guide KV selection—is intuitive and practically useful.

But for the drawback, the novelty is somewhat incremental relative to recent KV-cache sparsification and speculative decoding work (e.g., MagicDec, Quest, StreamingLLM). Besides, The paper lacks deeper analysis of why verification-derived attention signals reliably predict future KV importance. Also, the evaluation of this paper focuses on throughput; additional experiments on quality robustness or extreme context lengths should also included to show if this method works well in each metrics.

---

> ### Author Rebuttal · Authors · 2026-03-31
>
> # Q1: Novelty
> Along with the concurrent work SparseSpec, we are among the first to study verification-guided sparse attention in self-speculative decoding. While the intuition seems natural, we developed Vegas with the following new insights:
>
> (1) The attention weight collection will become a significant overhead that can undermine the performance benefit brought by better drafting accuracy, especially for LLM architectures like gpt-oss. This critical issue remains unsolved in SparseSpec, as they only evaluated Qwen3 models, which suffer less from this overhead but are still significantly affected (11% in overall decoding throughput for Qwen3-8B + AIME25). We tested vLLM + SparseSpec with gpt-oss-20b on AIME25 and found that its decoding throughput is 4% lower than MagicDec-Stream, as its collect-all-query strategy not only enlarges the attention weight collection overhead but also significantly reduces the number of available KV cache entries (see Q2 of Reviewer MbP4).
>
> (2) The important KV entries selected across draft tokens overlap heavily, allowing us to collect attention signals from only a subset of draft tokens. A natural choice would be the last few draft tokens, as they are closest to the upcoming draft tokens. However, we found that this selection becomes biased toward local attention patterns, providing little additional signal beyond the last draft token alone. Counterintuitively, including the first draft token — the one farthest from the tokens being drafted — turns out to be essential, as the two boundary tokens have the maximum positional distance and can effectively capture the diversity across all draft tokens. This collect-2-query approach significantly reduces the attention weight collection overhead and is essential for unleashing the full performance benefits of verification-guided sparse attention.
>
> # Q2: Stability of Verification Signals
> To evaluate the stability of verification signals, we measured the percentage of attention weight covered by Vegas's selected tokens over subsequent drafting steps, using StreamingLLM as a reference. We conducted the experiments with Qwen3-8B on LongBench-v2 at 7% sparsity ratio and γ = 7. Vegas consistently achieves higher attention weight coverage than StreamingLLM, demonstrating that verification-guided signals reliably retrieve important tokens beyond the local sliding window.
>
> |Drafting_step|1|2|3|4|5|6|7|
> |---|---|---|---|---|---|---|---|
> |StreamingLLM|59.8|59.5|59.7|59.8|59.9|60.5|60.3|
> |Vegas|77.2|77.3|78.0|78.3|79.0|80.0|80.9|
>
> # Q3: Generation Quality
> Vegas is a lossless acceleration method. Its outputs have the same distribution as vanilla decoding, as guaranteed by Speculative Sampling [1]. Thus, it has the same generation quality as vLLM. We report the number of correct answers of Qwen3-8B on AIME25 (30 problems in total), served by vLLM and Vegas (with the same setup as in Section 5.2), as follows. Vegas achieves accuracy comparable to vLLM across seeds 1-5, confirming that it preserves generation quality as theoretically guaranteed.
>
> |Generation Seed|1|2|3|4|5|
> |---|---|---|---|---|---|
> |vLLM|21|19|22|22|21|
> |Vegas|22|20|21|20|22|
>
> [1] Fast Inference from Transformers via Speculative Decoding
>
> # Q4: Extreme Context Length
> We tested Vegas’s performance with 1M context length, using the recently released Qwen3.5 models [2]. We integrated Vegas into vLLM v0.18.0 for this experiment. We used the LongCodeBench [3] dataset and selected requests with input lengths of 960K to 1M. With a batch size of 2 and a 7% sparsity ratio, Vegas achieves a 28% improvement in decoding throughput over MagicDec-Stream. This is because Vegas achieves a higher drafting accuracy (e.g., with 9 tokens drafted per iteration, Vegas has 7.13 accepted on average vs. 5.21 for StreamingLLM), while incurring minimal KV selection overhead.
>
> ||vLLM|MagicDec-Stream|Vegas|
> |---|---|---|---|
> |token/s|50.6|118.5|151.7|
> |γ|/|5|9|
>
> [2] Qwen3.5: Towards Native Multimodal Agents
>
> [3] LongCodeBench: Evaluating Coding LLMs at 1M Context Windows
>
> # Q5: Sensitivity to Sparsity Ratio and γ
> We provide a sensitivity analysis of the two parameters in Figure 12. Even if a sub-optimal configuration is selected (e.g., sparsity ratio = 5% and γ = 3), the achieved throughput of Vegas (218.0 token/s, as compared to the optimal 269.2 token/s with sparsity ratio 9% and γ = 9) still exceeds MagicDec-Stream at its optimal configuration (208.0 token/s). This is because Vegas achieves much higher drafting accuracy than StreamingLLM at 100K context (e.g., with 7% sparsity ratio and 11 tokens drafted per iteration, Vegas has 8.86 accepted on average vs. 5.89 for StreamingLLM), while incurring minimal KV selection overhead.
>
> # Q6: Impact statement
> Thanks for your suggestion. We will extend our Impact Statement to address broader implications of more scalable LLM deployment, including potential risks of LLM misuse.

---

> > ### Author Rebuttal · Reviewer_V1N7 · 2026-04-03
> >
> > I thank the authors for the detailed rebuttal and the additional experimental results.
> > The clarifications help better understand the implementation details and empirical behavior of the proposed method.
> >
> > However, the rebuttal does not directly address my primary concern, namely the underlying justification for why verification-derived attention signals can reliably predict future KV importance across decoding steps.
> > While the authors provide coverage statistics and empirical comparisons, these results mainly demonstrate performance improvements rather than explaining or validating the core assumption of the method.
> > In particular, the stability and generalization of these signals remain insufficiently analyzed from a mechanistic or theoretical perspective.
> >
> > Additionally, the sensitivity analysis and extended experiments, although useful, do not fully resolve the question of robustness under varying configurations or different workload regimes.
> > The evidence provided is still limited in establishing that the proposed approach consistently behaves as expected beyond the reported settings.
> >
> > Overall, while I appreciate the additional effort in the rebuttal, the main concerns regarding the justification of the key design choice and its robustness are not fully resolved.
> > Therefore, my overall assessment and recommendation remain unchanged.

---

> > > ### Author Response · Authors · 2026-04-08
> > >
> > > # Q1. Core Assumptions and Reasoning
> > >
> > > In Vegas, we had the following insights and verified each of them.
> > >
> > > ## 1. KV entries critical to the draft tokens under verification will remain critical for future drafting steps.
> > > To understand this insight, we studied the acceptance probability of the rejected tokens (Figure 5) and found that they actually have a high acceptance probability. Thus, these rejected tokens still align well with the context, and the critical KV entries they select remain useful, which is further confirmed by our ablation study (Line 213, right column).
> > >
> > > ## 2. Critical KV entries can be accurately identified based on a subset of draft tokens.
> > > To reduce the overhead of collecting attention logits, we assumed that critical KV entries can still be accurately identified using only the first draft token and the bonus token. To verify this, we compared drafting accuracy when predictions were based on all draft tokens vs. only the two tokens (Table 1). To explain this, we measured the overlap of critical tokens selected by each draft token and found high overlap (Figure 7).
> > >
> > > ## 3. Attention logits can be used to select critical tokens.
> > > To avoid the overhead of rematerializing attention weights, we assumed that using attention logit can achieve similar accuracy and empirically verified this (Line 700, left column). In our response to Reviewer JwrV, we further verified that this is because the cancellation effect (i.e., a token with high attention weights is not selected because a few extremely low logit entries) is not severe, due to the high correlation of the output across query heads.
> > >
> > >
> > > # Q2. Robustness Justification
> > >
> > > ## Configurations
> > >
> > > Vegas’s performance is related to two configurations: (1) the sparsity ratio and (2) the number of tokens drafted per decoding iteration (i.e., γ).
> > >
> > > Vegas's performance is robust to the sparsity ratio: as long as a reasonable sparsity ratio is set (e.g., 5%–7%), Vegas delivers reasonable performance improvement. This is because Vegas’s drafting accuracy typically stabilizes once the sparsity ratio exceeds 5% (Figure 8), and it is much higher than our baselines (e.g., in Figure 10(a), 34% higher than MagicDec-Stream). Further increasing the sparsity ratio within a reasonable range (e.g., <10%) slows drafting slightly and has a minor impact on overall decoding throughput (Figure 12).
> > >
> > > Vegas's performance is also robust to γ, with a reasonable sparsity ratio such as 7%. For example, in comparison to MagicDec-Stream with its best configuration, Vegas achieves at least 7% improvement for γ ranging from 3 to 11 when serving Qwen3-8B on LongBench-v2 (Figure 12). With an even smaller γ (e.g., 1), Vegas’s performance is similar to that of MagicDec-Stream because of its minimal KV selection overhead. With a larger γ (e.g., 20), although Vegas may not offer benefits over vanilla decoding, its performance is still better than the speculative decoding baselines, as it can achieve higher drafting accuracy. To further improve robustness, it is practical to dynamically configure the optimal γ in Vegas, following a similar approach explored in [1].
> > >
> > > [1] SpecDec++: Boosting Speculative Decoding via Adaptive Candidate Lengths (COLM’25).
> > >
> > >
> > > ## Workloads
> > >
> > > Vegas is designed for long-context serving, as reasoning models and agentic LLM applications are becoming pervasive. When serving long contexts, Vegas’s performance is robust in both throughput- and latency-oriented scenarios.
> > >
> > > For throughput-oriented scenarios, Vegas achieves a 1.25 – 2.7× speedup over vLLM and 1.15 – 1.23x over MagicDec-Stream (Figure 9) with ~10K contexts, and the speedup is higher for ~100K contexts (Figure 11), because a longer context makes the critical token selection more challenging while Vegas maintains high drafting accuracy (Figure 8). For even longer contexts, such as 1M, Vegas achieves 28% speedup over MagicDec (your Question 4).
> > >
> > > For latency-oriented scenarios, Vegas’s performance remains robust: Vegas begins to deliver a 20% performance gain over vLLM at a batch size of 8 for 10K contexts and at a batch size of 1 for 100K contexts (see Q3 of Reviewer MbP4). This is because Vegas achieves high drafting accuracy with minimal KV selection overhead. Even in cases such as batch size 1 with a 10K context length, Vegas still matches the performance of the baselines for the same reason.
> > >
> > > Note that in short-context scenarios, all sparse-attention-based speculative decoding methods suffer from performance degradation. This is because, if the batch size is small, the overall computation will be dominated by FFN, which makes drafting with sparse KV cache not beneficial. If the batch size is large, then FFN computation in both drafting and verification will become compute-bound. Each additional drafted token significantly increases verification latency. For such scenarios, Vegas delivers comparable performance to state-of-the-art sparse attention-based solutions, as it causes minimal KV selection overhead.

---

### Official Review · Reviewer_MbP4 · 2026-03-12

**Soundness:** 3
**Presentation:** 3
**Significance:** 2
**Originality:** 3
**Overall Recommendation:** 4
**Confidence:** 3

**Summary:**

The paper proposes Vegas, a self-speculative decoding method that co-designs drafting and verification to accelerate long-context LLM inference under KV-cache bandwidth constraints. During verification, Vegas collects attention logits and uses them to select a small set of “critical” KV entries, which are then used for sparse attention in subsequent drafting steps. To keep overhead low, Vegas aggregates logits only from the first draft token and the bonus token (“Collect-2-Query”), and it provides a simple hyperparameter tuning strategy over the sparsity ratio and number of drafted tokens. Integrated into vLLM and evaluated on Qwen3 and gpt-oss-20b models across AIME25, CodeElo, and LongBench-v2, Vegas achieves 1.25×–2.81× throughput over vanilla decoding and 1.18×–1.29× improvements over strong sparse self-speculative baselines.

**Compliance With Llm Reviewing Policy:**

Affirmed.

**Final Justification:**

Resolved my concern but still a bit worried about the novelty.

**Key Questions For Authors:**

1. **Baseline hyperparameters:** How were sparse ratios, window sizes, and other relevant hyperparameters for MagicDec-Quest and SpecExtend tuned on each model and dataset? Were they subject to a similar three-stage sweep as Vegas, or were defaults used? Clarifying this, ideally with a short table of search ranges, would help establish fairness of comparisons.

2. **Memory overhead:** Can you quantify the exact additional HBM footprint of Collect-2-Query (per layer per batch) at 120K context length and maximum batch size? Does this reduce the maximum batch that fits on an H100 compared to baselines, and if so, by how much?

3. **Small-batch latency behavior:** Have you profiled Vegas at small batch sizes (e.g., batch = 1–4) on any of the benchmarks? Does it still provide net latency reduction over vLLM and MagicDec-Stream, or does overhead dominate? A brief result or clarification in the paper would be helpful.

**Limitations:**

1. **Novelty is mostly in engineering optimization rather than a fundamentally new idea:** The core conceptual step—using verification attention to guide drafting sparsity—is natural given existing self-speculative frameworks and is also present in concurrent works like SparseSpec and the uncited SpecAttn variants. Vegas’s unique pieces (logits vs weights, Collect-2-Query) are clever but incremental. For an ICML main-track paper, this is acceptable but not standout.

2. **Baseline ecosystem is incomplete, which blurs the magnitude of the contribution:** While MagicDec and SpecExtend are strong baselines, several highly relevant speculative/sparse decoding schemes are missing (e.g., sparse verification, QuantSpec, other verification-guided KV approaches). Without at least a conceptual comparison and, ideally, some empirical numbers, it is hard to fully assess Vegas’s relative standing in the rapidly evolving LLM inference landscape.

3. **Evaluation lacks small-batch, low-latency scenarios:** All throughput numbers are measured with relatively large batches aimed at saturating HBM (e.g., up to batch size 128 on H100). Many real-world deployments serve interactive queries at small batch sizes where per-request latency, not throughput, is paramount. It is conceivable that the overhead of Collect-2-Query and speculative control logic offers less benefit or even hurts latency in that regime. The paper does not explore this, so the practical envelope of Vegas remains somewhat unclear.

**Strengths And Weaknesses:**

1. **Clear co-design insight with concrete implementation:** The core idea of reusing verification attention information to guide subsequent sparse drafting is well-motivated and concretely realized. Figure 1 nicely contrasts the baseline speculative framework vs. Vegas and makes the pipeline-level change easy to understand.

2. **Technically coherent sparse KV selection:** The KV selection strategy, formalized in Equations (2) and (3), aggregates attention logits across multiple draft tokens and heads, which empirically avoids overfitting to the last accepted token. Figure 4 clearly shows that the per-position acceptance rate stays high across the draft chain under the proposed multi-token coverage objective, in contrast to the rapid decay when using only the last accepted token.

3. **Careful optimization of overhead via Collect-2-Query:** The Collect-2-Query design is a strong systems contribution. Figure 6 convincingly shows that collecting logits for all draft tokens dominates FA3’s latency, while restricting collection to just the first draft and bonus tokens essentially flattens the overhead curve, yet Table 1 shows almost identical accepted-token counts.

---

> ### Author Rebuttal · Authors · 2026-03-31
>
> # Q1: Baseline Hyperparameters
> We did sweep for the best configuration for each baseline. First, we examined sparsity ratios of 1%, 3%, 5%, 7%, and 9% for all models on LongBench-v2 and selected 7% (Step 1). Next, we swept γ from 1 to 11 for each method and workload; the selected γ values are shown in Table 2 (Step 2). Finally, we confirmed that further tuning of the sparsity ratio does not have a significant impact on the decoding throughput (<2%) (Step 3). Thus, for a fair comparison, all reported results use a sparsity ratio of 7%.
>
> vLLM manages KV cache in pages. Thus, for Quest and SpecExtend, we set their page size to vLLM's default value of 16. For StreamingLLM, we selected the last ⌈sparsity_ratio * context_len / 16⌉ pages as the sliding window. Additionally, we include the first page of each request for StreamingLLM as the attention sink.
>
> # Q2: Memory Overhead
> The HBM footprint of Vegas is dominated by the buffer (>95%) used to collect attention logits in BF16, with shape [max_batch_size, num_q_heads, 2, max_context_len] (2 for collect-2-query). This buffer is reused across layers.
>
> In our evaluation, the largest buffer size is 4 GB for gpt-oss-20b + AIME25/CodeElo: 128 (batch size) × 64 (query heads) × 2 × 128K (context length) × 2 (bytes per BF16) = 4 GB. For other models in our experiments, the overhead is <1 GB.
>
> Vegas’s collect-2-query design significantly reduces the memory overhead compared to the naive collect-all-query method. With vLLM’s default setting of 90% HBM utilization, 67.8 GB is allocated for KV cache. The naive collect-all-query approach requires a 12 GB buffer (at γ = 5) and reduces the available KV cache capacity by 12 / 67.8 = 17.7%. In contrast, Vegas incurs at most 5.9% capacity overhead. The negative impact on decoding throughput is fully offset by the benefits of Vegas.
>
> # Q3: Small Batch Sizes
> We evaluated batch sizes 1, 2, 4, and 8 with Qwen3-8b + AIME25; the sparsity ratio is set to 7%. We report the decoding throughput and time-per-output-token (TPOT) latency of vLLM, MagicDec-Stream, and Vegas as follows:
> |Batch_size|1|2|4|8|
> |---|---|---|---|---|
> |**Throughput (token/s)**||||
> |vLLM|130|239|400|587|
> |MagicDec-Stream|121|208|390|670|
> |Vegas|119|210|399|702|
> |**TPOT (ms)**|||||
> |vLLM|7.69|8.37|10.00|13.63|
> |MagicDec-Stream|8.26|9.62|10.26|11.94|
> |Vegas|8.40|9.52|10.03|11.40|
> |**γ**|||||
> |MagicDec-Stream|1|3|4|4|
> |Vegas|1|3|4|4|
>
> Vegas works best for throughput-oriented workloads and long-context serving, where the KV cache loading time is the major bottleneck. This is because drafting with sparse attention in self-speculative decoding only accelerates the attention layers. For very small batch sizes with relatively short context lengths, the FFN computation time dominates, and Vegas can be 8%-12% slower than vLLM at batch sizes of 1 or 2. MagicDec-Stream suffers from the same overhead at small batch sizes. At a batch size of 8, the attention time outweighs the FFN time, and Vegas achieves 19.6% higher throughput than vLLM.
>
> For long-context requests (e.g., 100K tokens), KV cache loading time is significant even for small batches. In such cases, Vegas achieves a TPOT of 9.82 ms, 20% lower than vLLM's 12.3 ms at batch size 1 (Qwen3-8B + LongBench-v2, γ = 7).
>
> # Q4: Novelty and Contribution
> Please see Q1 of Reviewer V1N7.
>
> # Q5: Related Work
> Sparse verification [1] sparsifies attention, MoE, and FFN computation to accelerate the verification phase. It trades generation quality for speed and does not co-design drafting and verification. In contrast, Vegas is a lossless acceleration method.
>
> SpecAttn [2] uses the **drafter** model's attention pattern to select KV entries for the sparse attention computation in the **target** model. It also trades generation quality for speed. In contrast, Vegas employs sparse attention for drafting tokens.
>
> QuantSpec [3] achieves self-speculative decoding by employing a quantized target model with a quantized KV cache as the drafter. Vegas achieves a much higher KV compression ratio (it only needs to read 5-7% of full KV cache size) than QuantSpec's 4-bit method (25%) while maintaining comparable drafting accuracy. Additionally, QuantSpec requires the KV cache to be stored in INT8, whereas Vegas does not enforce a specific data type. Regarding weight quantization, we believe that combining QuantSpec's weight compression technique with Vegas's sparse attention could provide further speedups.
> We will incorporate the above discussion into our paper.
>
> [1] Accelerate Speculative Decoding with Sparse Computation in Verification
>
> [2] SpecAttn: Speculating Sparse Attention
>
> [3] QuantSpec: Self-Speculative Decoding with Hierarchical Quantized KV Cache

---

> > ### Author Rebuttal · Reviewer_MbP4 · 2026-04-02
> >
> > Thanks for the response. I have raised the score.

---

### Official Review · Reviewer_sgBq · 2026-03-13

**Soundness:** 3
**Presentation:** 3
**Significance:** 3
**Originality:** 3
**Overall Recommendation:** 5
**Confidence:** 4

**Summary:**

This paper proposes Vegas, a self-speculative decoding method for long-context LLM inference that co-designs drafting and verification. The key insight is that verification already computes full attention and thereby knows the exact criticality of every KV entry; Vegas reuses this information to guide sparse attention in the subsequent drafting phase. To keep overhead low, a "Collect-2-Query" mechanism extracts attention logits from only the first draft token and the bonus token rather than all draft tokens. Vegas is implemented within vLLM as a training-free system and evaluated on short-input reasoning (AIME25, CodeElo) and long-context (LongBench-v2) benchmarks across four models, reporting 1.25–2.81× throughput improvement over vanilla vLLM and 1.18–1.29× over the strongest sparse self-speculative baselines.

**Compliance With Llm Reviewing Policy:**

Affirmed.

**Final Justification:**

This paper makes a solid systems contribution to efficient long-context decoding, with a simple idea and strong empirical gains. My main concerns were missing experimental details and questions about the practical serving setup. The authors’ rebuttal addressed these well. My remaining reservation is that the novelty is stronger on the systems side than at the high-level algorithmic level, but overall, the paper is meaningful and well supported. The rebuttal increased my confidence, and I raised my score.

**Key Questions For Authors:**

1. **Figure 8 setup.** What dataset, context length distribution, and batch size were used to generate Figure 8?
2. **Page size tradeoff.** The implementation uses page size 1 for token-level sparsity. Did the authors evaluate larger page sizes, and how does coarser selection granularity trade off against reduced indexing overhead and better memory locality?
3. **Batch occupancy and heterogeneous batch states**. With a maximum batch size of 128 and a small number of distinct AIME25 problems (for example, for AIME25), are requests replayed to max maintain occupancy? If so, newly admitted requests will have short KV caches while in-flight requests may have accumulated thousands of tokens. Also, new requests can be in a different stage (draft or verify) than in-flight requests. This can result in heterogeneous batch states where different requests are simultaneously at different speculative stages. How does Vegas handle this, and how does this affect the stable throughput reported?
5. **Short-prefix regime on AIME25.** With average input length of only 182 tokens, the KV cache is small early in generation and the cost advantage of sparse drafting is modest. Does the reported stable throughput reflect a steady state reached only after substantial generation, and how does Vegas's advantage evolve over the course of a typical request?

**Limitations:**

yes

**Strengths And Weaknesses:**

# Strengths
1. **Clean and well-motivated core insight.** Reusing verification's full-attention computation as a "free oracle" for KV selection is simple,
well-supported, and directly addresses the drafting-accuracy/overhead tradeoff.
2. **Collect-2-Query is the paper's strongest technical contribution.** Reducing collection overhead from 53–189% (collect-all) to 5–37% while preserving drafting accuracy is a meaningful and practical result, well-supported by Figure 6 and Table 1.
3. **Efficient and practical implementation.** Vegas integrates cleanly into vLLM by intercepting calls to the existing attention dispatch path and instrumenting the FlashAttention-3 kernel to write attention logits to HBM with minimal added overhead. This hook-based design avoids modifications to the core serving framework and makes the system immediately deployable on existing infrastructure.
4. **Systematic empirical methodology.** The paper isolates and quantifies the effect of each key design choice: drafting accuracy, logit collection overhead, the value of including rejected tokens, and sensitivity to both sparse ratio and draft length. This makes it straightforward to understand where each component of the throughput gain originates and how to tune the system for a new model or workload.
5. **Strong empirical gains in the long-context regime.** Vegas's 18–29% throughput advantage over the strongest competitor on LongBench-v2, and the scaling argument are persuasive.

# Weaknesses
1. **"Plug-and-play" is overstated.** Achieving reported gains requires sweeping both sparse ratio and draft length per model and dataset (Table 2 shows substantial variation). This non-trivial tuning cost sits in tension with the plug-and-play framing.
2. **Incomplete experimental specifications.** The setup for Figure 8 is not stated in the main text. The two-GPU prefill-decoding disaggregation used for LongBench-v2 is mentioned but its implications for the throughput numbers are not discussed.
3. **Batching and scheduling underexplained.** The paper does not explain how Vegas handles heterogeneous batch states (requests at different speculative stages) within vLLM, making it difficult to assess how results generalize to real serving workloads.
4. **Model name inconsistency.** The a introduction reference "gpt-oss-120b" while the evaluation consistently uses "gpt-oss-20b."

---

> ### Author Rebuttal · Authors · 2026-03-31
>
> # Q1: Plug-and-play
> We wish to clarify that "plug-and-play" means Vegas does not require training a draft model, and is ready to be plugged into vLLM as a drop-in module. We will clarify this in the paper.
>
> In practice, users can reduce the tuning time based on their needs: the sparsity ratio can be empirically set to 5%–7%, and only γ needs to be tuned if necessary, as shown in Figure 12.
> # Q2: Figure 8 Setup
> Figure 8 uses LongBench-v2 with the same setup as in Section 5.3. The input context lengths range from 96K to 120K. Batch size is 4 for Qwen3-8B and 20 for gpt-oss-20b. Batch sizes do not affect the drafting accuracy.
> # Q3: Prefill-Decode Disaggregation (PDD)
> We employed PDD since it provides better end-to-end throughput for Vegas and all speculative decoding baselines. Besides, PDD isolates decoding performance from prefill interference and provides a clearer breakdown of benefits. We report the decoding throughput (Qwen3-8B, the same setup as in Section 5.3) without PDD below:
> ||vLLM|MagicDec-Stream|MagicDec-Quest|SpecExtend|Vegas|
> |---|---|---|---|---|---|
> |token/s|67.2|81.3|78.2|75.9|98.4|
>
> Without PDD, the decoding throughput is lower due to prefill interference, but Vegas still outperforms the strongest baseline by 1.21×.
> # Q4: Typo
> Thanks for pointing out the typo. We will correct the “gpt-oss-120b” typo in the paper. All our experiments use gpt-oss-20b.
> # Q5: Page Size Tradeoff
> We benchmarked the FlashAttention-3 (FA-3) kernel with various page sizes on H100 GPUs. We set batch size to 16 (to amortize kernel launch overhead), allocate KV buffer for 16*128K tokens, and select 16*8192 tokens (each request has 8192/page_size pages) with page positions randomly chosen from the full buffer. We report the FA-3 kernel latency (in μs) below:
> |Page Size|1|2|4|8|16|
> |---|---|---|---|---|---|
> |Qwen3-8B|197.6|197.6|197.1|198.1|198.3|
> |gpt-oss-20b|131.0|130.1|130.1|130.3|130.8|
>
> The latency is almost identical for page sizes 1 and 16 (vLLM's default). The differences are negligible because Vegas selects the same tokens for all KV heads. With the NHD KV layout (the tensor shape is [num_pages, page_size, num_kv_heads, head_dim]), the KV cache per token forms a contiguous buffer of 4KB and 2KB for the two setups, respectively, which already provides sufficient memory locality.
>
> Larger page sizes may reduce TopK kernel latency and page table size. However, the savings are negligible, since the values of both the TopK latency and the page table size are trivial. Hence, we use token-level selection to maximize the drafting accuracy.
> # Q6: Heterogeneous Batch States
> In Vegas, there are no "heterogeneous batch states". This is because the scheduling policy of vLLM v1 runner (which Vegas is built upon) ensures that all in-flight requests are simultaneously in either the verification or drafting phase (the γ drafting steps are also aligned).
> # Q7: Batch Occupancy at Steady State
> The 30 distinct questions in AIME25 are replayed 32 times. The decoding throughput stabilized after ~400 requests had been served, at which point we began measuring. The default max context length is used: 40K for the Qwen3 family and 128K for gpt-oss-20b.
>
> At steady state, in-flight requests have varying context lengths, as shown in the CDF below (Qwen3-8B + AIME25, 10000 decoding steps sampled):
> |Percentile|1|5|10|25|50|75|90|95|99|
> |---|---|---|---|---|---|---|---|---|---|
> |Context length|292|880|1515|3818|9507|17727|25232|29438|37376|
>
> # Q8: Short-Context Regime
> Vegas ensures that each request selects at least 128 tokens (min_budget=128) during sparse attention. While sparse drafting may not be beneficial for requests with very short context (e.g., <256 tokens), this does not significantly impact the overall decoding throughput because such short contexts typically account for <1% at steady state (see the CDF in Q7).
>
> Specifically, sparse drafting is beneficial when the amortized KV traffic per generated token is lower than reading the full KV cache. Let $s$ denote the actual sparsity ratio (e.g., if 128 tokens are selected from 160 tokens, then $s$=80%) and γ denote the number of drafting steps per decoding iteration. The full KV cache is loaded (γ×$s$+1) times to generate (γ×acceptance_rate+1) tokens. Thus, per generated token, the full KV cache is loaded $r$ = (γ×$s$+1) / (γ×acceptance_rate+1) times.
>
> For a typical request, if $r<1$ (i.e., $s$ < acceptance_rate), sparse drafting is beneficial. Consequently, when the context length is shorter than min_budget/acceptance_rate (e.g., with a conservative acceptance rate of 50%, the threshold is 128/0.5=256 tokens), we have $r>1$, so the cost of sparse drafting outweighs the benefit. Beyond this threshold, the benefit outweighs the cost, and the gap grows and stabilizes once the context length reaches min_budget/target_sparsity_ratio (e.g., this threshold is 1829 with 7% sparsity ratio). Beyond this point, only the average acceptance rate impacts the performance benefit.

---

> > ### Author Rebuttal · Reviewer_sgBq · 2026-04-03
> >
> > Thank you for the clear and helpful response. It addressed my concerns and clarified several important details. I have raised my score.

---

### Official Review · Reviewer_JwrV · 2026-03-13

**Soundness:** 2
**Presentation:** 3
**Significance:** 3
**Originality:** 3
**Overall Recommendation:** 5
**Confidence:** 3

**Summary:**

The authors speed-up speculative decoding in transformer LLMs by improving upon previous work that uses a transformer with sparse attention as the draft model and the same transformer with full-attention as the verifier. The authors find a better trade-off between drafting accuracy and KV selection overhead than previous work: they use information from the verifier's full KV attention to pick which KV-cache entries to drop for the drafter at the next step. Their strategy simplifies to picking the k tokens in the context window that maximize the sum of the attention logits (as computed by the verifier) for the first token in the draft window and the bonus token in the draft window. They show that they achieve better acceptance rates while keeping the introduced latency from sparsifying the KV cache lower than alternative baselines that use the same sparse attention speculative decoding setup. As a result, they speed up generation for gpt-oss-20b by at least 1.25x over vLLM and up to 2.7x for the 4B Qwen models, with improvements over the best alternative that uses sparse attention (MagicDec-Stream) being between 1.15x and 1.23x in throughput.

**Compliance With Llm Reviewing Policy:**

Affirmed.

**Final Justification:**

The authors have addressed all my concerns and I appreciate their detailed response, clarification about the effect of relative differences in logits and the additional empirical evidence on 4 more models which shows that using the logits leads to similar acceptance rates as using the attention weights. I have increased my score accordingly.

**Key Questions For Authors:**

1. Not all tasks require huge context windows, Vegas's strength seems to be in the long context case, but could the authors comment on how Vegas performs in short contexts?
2. What is the number of critical entries in a KV-cache that need to be included in sparse attention and does it necessarily grow as context increases? Can the sparsity ratio the authors define be a fixed number that is independent of the context length?
3. The authors write on line 212: "This is because selecting KV entries based on a single token may overfit to that token's specific attention pattern". Since the authors are not fitting a model to select the KV entries, what does overfitting mean in this context? The authors also use the term overfitting in section 3.3 in a way that seems to mean "reduces the acceptance rates". Could the authors directly discuss the effect of their choices without using the term overfitting which seems a bit distracting? It would be great if they could elaborate why they chose that term.
4. In 3.3, could the authors please clarify what the overhead of collecting attention logits is measured in comparison to? Is it speculative decoding with a fixed sparsification criterion?

**Limitations:**

It would be great if the authors could elaborate on any potential limitations of their method on shorter contexts and the point about the sign of logits and their magnitude impacting which KV-cache entries are retained.

**Strengths And Weaknesses:**

## Strengths

1. (**Presentation / Significance**) The work is relevant to the ICML community, the contribution is timely and on an important problem, the paper is nicely presented, well structured and easy to follow, the experimental setup is sensible, and the authors show that they can outperform reasonable baselines by 1.15x - 1.23x in throughput.
2. (**Soundness / Significance**) The authors provide evidence and ablations for many of the choices they made, e.g. picking KV entries with highest attention weights for the last accepted token vs all draft tokens vs the first token and bonus token (Figure 4 and Table 1), and the fact that the attention weights from rejected tokens can still be useful (Figure 5). These ablations provide additional insights into how to select which entries of the KV cache to keep, a problem that may be useful for follow-up work.
3. (**Originality**) The paper is insightful and contains some nice observations on how to make this speculative decoding setup more efficient.

## Weaknesses

1. (**Soundness**) While the authors empirically demonstrate that computing the argmax over the sum of attention logits rather than the normalised attention activations works for their models, can we be certain that this finding generalises across models? It is a bit unclear what happens when summing logits, since logits can be both positive and negative and there may be cancellation effects at play that could severely change the result compared to the sum of normalised attention activations. Do the authors have any insights on why this works and did they investigate what happens with a few examples? It would be useful if they could provide more evidence that this is a reasonable idea, or alternatively, provide a few checks for when things can go wrong. It would also be useful if the authors could do a more in-depth analysis by looking at the sign of the logits and their magnitudes and comparing to the corresponding normalised attention activations.
2. (**Presentation**) While the presentation is generally clear, some parts can be improved. In the abstract, it was a bit unclear what criticality is. Moreover, the authors are not explicit early on in the paper about the temperature they use with speculative sampling (they mention sampling on page 4). It would be useful to the reader if the authors mention this when they discuss rejecting tokens. Lastly, three tokens are of particular interest for the method and ablations: the first draft token, the last accepted token and the bonus token. It would be useful for the reader if these were introduced together after discussing the draft token window and/or if these appeared in the plot discussing the method.

## Comments

Figure 7 takes up a lot of real estate in the paper and seems to not advance the story much. Table 1 already seems like a good enough justification for the 1st + bonus token?

In Figure 5, is that the acceptance probability rather than the acceptance rate on the x-axis?

## Typos

* sparse ratio -> sparsity ratio
* equation (1), and elsewhere q -> q^\top (q is a row vector? please elaborate on the notation used.)
* gpt-oss-120b -> gpt-oss-20b

---

> ### Author Rebuttal · Authors · 2026-03-31
>
> # Q1: Attention Logit vs Weight
> We observed that tokens with high attention weights across multiple heads rarely have attention logits with large negative values, so the 'cancellation effect' does not significantly impact our logit-based KV selection. For Qwen3-8B/gpt-oss-20b + LongBench-v2, among the top 5% of tokens ranked by attention weights, <9% have an individual attention logit drop below two standard deviations from the mean within the attention head. Furthermore, we verified that tokens with the top 5% average attention weights can be recalled by 84% on average using the top-5% tokens ranked by attention logits.
>
> While we empirically verified this phenomenon specifically on Qwen3 and gpt-oss-20b, we expect this to generalize to many other models due to the inherent redundancy across attention heads [1]: if a specific head produces attention logits with large absolute values (either positive or negative), other heads likely produce similar values. Hence, it is rare for a token to be overlooked by attention logit ranking solely due to an extremely low logit from a single head.
>
> [1] Are Sixteen Heads Really Better than One?
> # Q2: Suggested Presentation Improvements
> Thanks for your suggestions. We will take an editorial pass of the paper to further improve its presentation quality.
> (1) In the Abstract, we will highlight the trade-off between drafting accuracy and KV selection overhead in existing work; Vegas breaks this trade-off with verification-guided KV selection.
> (2) We will introduce the sampling temperature along with speculative sampling. We will also report all sampling parameters used, which follow the defaults recommended on HuggingFace.
> (3) We will introduce the last accepted token in Section 2.3. We will also add an illustration to further clarify the first/last draft token and the last accepted token.
> # Q3: Functionality of Figure 7
> We would like to keep Figure 7, as it provides two intuitions: (1) the high overlap among critical KV subsets identified by each draft token motivates our design of collecting attention logits from a subset of draft tokens, and (2) the overlap decreases with a larger positional distance between draft tokens, which explains our choice of collect-2-query.
> # Q4: X-axis of Figure 5
> Thanks for pointing this out. The x-axis of Figure 5 and the y-axis of Figure 4 should both be “acceptance probability”. We will correct them.
> # Q5: Typos
> We will correct the typos in the paper:
> (1) “Sparse ratio” should be “sparsity ratio”.
> (2) The shape of q in Section 2.1 is 1×d.
> (3) “gpt-oss-120b” should be “gpt-oss-20b”.
> # Q6: Short Contexts
> For short contexts, the maximum concurrent batch size is large enough to saturate GPU compute (e.g., the batch size can reach 240 for Qwen3-8B and 1200 for gpt-oss-20b at 2K context length), making FFN computation in both drafting and verification compute-bound. Each additional drafted token significantly increases verification latency; in such a scenario, speculative decoding is not beneficial.
> # Q7: Sparsity Ratio vs. Fixed Budget
> To maintain drafting accuracy, more tokens need to be selected for sparse attention as context length grows. We tested two strategies for determining the token budget: (1) a fixed count of 1,536 tokens (~7% of the average context length of 19,862 tokens) and (2) a 7% sparsity ratio, on Qwen3-8B+AIME25. We report the average number of accepted tokens per decoding iteration at varying context lengths with 6 tokens drafted per decoding iteration:
> |Context length|8K|16K|24K|32K|40K|
> |---|---|---|---|---|---|
> |1536 tokens|5.52|5.42|5.40|5.39|5.38|
> |7% tokens|5.23|5.30|5.45|5.58|5.60|
>
> With a fixed budget, drafting accuracy decreases as context length grows. In contrast, with a linearly growing budget, drafting accuracy increases slightly. The achieved decoding throughput is nearly the same for both strategies because the number of accepted draft tokens per iteration differs by <6%. Besides, the additional cost of choosing more tokens offsets the accuracy benefit.
>
> We use the sparsity ratio in our design for flexibility, as different requests may favor different budgets. For example, requests in AIME25 favor a budget around 1536 tokens, while requests in LongBench-v2 favor a budget around 5120 tokens.
> # Q8: Clarification on the Use of "Overfitting"
> When KV entries are selected based solely on a single token, the selection becomes biased toward (i.e., “overfits”) that specific token's attention pattern and cannot faithfully represent the attention pattern of the general local context. We will clarify this in the paper.
> # Q9: Baseline for Measuring Attention Logit Collection Overhead
> In Section 3.3, the attention logit collection overhead is relative to the attention kernel latency without attention logit collection. For example, for Qwen3-8B with num_query=12, the attention kernel takes 859.2 μs without attention logit collection and 1312.4 μs with collection, thus the overhead is 1312.4/859.2−1=53%.

---

> > ### Author Rebuttal · Reviewer_JwrV · 2026-04-02
> >
> > Thank you to the authors for their detailed response. I have a few follow-up questions.
> >
> > Q1: **Attention Logit vs Weight**: While I appreciate the motivation via the cited paper [1] and the additional statistics based on Qwen3-8B/gpt-oss-20b, I am not convinced that the provided evidence suffices for the general claim that using attention logits instead of weights will generalize to other models. I would encourage the authors to provide more evidence for their claim by reporting logit statistics from more models. More specifically, a) The reported deviation from the mean is not necessarily informative about the sign. Could the authors report statistics that directly support their claim? E.g. what percentage of logits is positive vs negative and what is the sum of positive logits and the sum of negative logits across heads? b) Could the authors elaborate on why they report additional statistics only for the top 5%? I believe this has to do with the sparsity ratio, but as far as I could tell they used a sparsity ratio of 7% in the paper (Appendix B). Since values up to 11% were used in the paper (figure 8), would it not make sense to report results for a larger threshold as well? c) Could the authors point out where their claim "if a specific head produces attention logits with large absolute values (either positive or negative), other heads likely produce similar values" is supported in [1]? d) the results are based on two models, how can the authors be certain that the logit statistics remain the same across models and setups?
> >
> > Q6 **Short Contexts**: I believe the authors assume that we are in a throughput-oriented workload scenario where when GPU compute is saturated, speculative decoding is not useful. But what about a latency-oriented scenario where speculative decoding would still be needed/useful, e.g. for models where autoregressive generation has higher latency than can be tolerated by end users? Could the authors discuss this distinction and guide the reader as to which scenarios their model is most effective for?
> >
> > Q8 **Overfitting**. I thank the authors for the clarification. Overfitting suggests fitting to noise on the training data in a way that hinders generalisation; this differs semantically from how they are using the term. Why is the term overfitting appropriate? How do the authors intend to clarify this in the paper?

---

> > > ### Author Response · Authors · 2026-04-08
> > >
> > > # Q1: Attention Logit vs. Weight
> > >
> > > ## Sign/sum of logits
> > > We appreciate your suggestions on logit statistics. We wish to clarify that statistics such as the percentage/sum of positive/negative logits are not relevant to the “cancellation effect”. This is because, given all attention logits from a query head, if we add the same arbitrary constant to every logit (i.e., a translation):
> > > 1. All attention weights remain unchanged: since $\text{softmax}(x) = \text{softmax}(x + c)$, attention weights are determined solely by the **relative differences** between logits.
> > > 2. The tokens selected by attention logits remain unchanged: since adding a constant $c$ to all numbers in a set does not change their sorted order, the same top $k$ tokens will always be selected regardless of the absolute values.
> > >
> > > Therefore, although the signs and sum of the attention logits change, the selection results of both methods remain unchanged. This means the "cancellation effect" remains unchanged under arbitrary logit translation and is independent of these statistics. The average logit varies across LLMs, our experiments show that it is -9.13 for Qwen3-8B and -139.0 for gpt-oss-20b.
> > >
> > > ## Deviation from the mean
> > > We clarify the reason for "deviation from the mean" reported in our prior response. The "cancellation effect" occurs when a token has a logit entry that ranks very low in one of the query heads. The absolute value is not a useful criterion for evaluating “cancellation effect”: for example, a -50 logit will not cause a “cancellation effect” if the average logit of that head is -100. Therefore, in our previous response, we reported how many of the top-weighted tokens have at least one low-ranking logit in any query head (i.e., more than 2 standard deviations below the mean; note that the deviation is relatively stable across heads).
> > >
> > > ## Results with additional LLMs
> > > Our study shows that attention logits are highly correlated across heads, making it rare for an overall important token to be “canceled” by a single low-ranking logit entry. Specifically, we find that:
> > >
> > > 1. The number of accepted tokens per iteration (7% sparsity ratio, γ=7, LongBench-v2) is similar under both logit- and score-based selection.
> > > 2. The percentage of top-11% weight-ranked tokens with any logit less than two standard deviations below the mean within the attention head remains <10%.
> > > 3. The Pearson correlation of attention logits between heads is typically >0.7 within the same GQA group and >0.5 across groups.
> > >
> > > We report our study results with mainstream LLMs as follows:
> > >
> > >
> > >
> > > |LLM|Accepted Tokens (weight/logit)|Percentage of Tokens with outlier logit (mean/std)|Cross-head Pearson correlation (same/different GQA group)|
> > > |---|---|---|---|
> > > |Llama-3.1-8B|5.69 / 5.54|5.65 / 4.39|0.84 / 0.51|
> > > |Nemotron-3-Nano-30B-A3B-BF16|6.14 / 6.15|2.24 / 2.07|0.83 / 0.75|
> > > |gemma-3-12b-it|6.01/5.99|4.46 / 3.27|0.80 / 0.55|
> > > |Ministral-3-14B-Reasoning|6.09/6.01|7.53 / 5.21|0.73 / 0.52|
> > >
> > >
> > > ## Only 5% sparsity ratio used
> > >
> > > We agree this is a valid concern; we report statistics on the top-11% tokens this time (see "Results with additional LLMs").
> > >
> > > ## Paper [1]
> > > Paper [1] claims that most attention heads can be redundant. We agree this does not directly imply high correlation among attention logits/weights produced by different heads on the same token. With a recent study [2] that reported “there is a high amount of correlation across the output of various attention heads in MHA”, and “the output of several attention heads focuses on the same token”, and our experiments with mainstream LLMs, we can reasonably explain our hypothesis that “if a specific head produces attention logits with large absolute values, other heads likely produce similar values”. We will add the discussion in the paper.
> > >
> > > [1] Are Sixteen Heads Really Better than One? (NeurIPS’19)
> > >
> > > [2] CHAI: Clustered Head Attention for Efficient LLM Inference (ICML’24)
> > >
> > >
> > > ## Generalizability
> > > Logit-based KV selection is translation invariant with respect to the logit distribution, making it less vulnerable to variations in LLM architectures. Our empirical study with diverse LLMs show that this algorithm generalizes across diverse LLM architectures – especially for the models that exhibit high inter-head output correlation.
> > >
> > > # Q2: Low-latency scenarios
> > >
> > > Vegas is less efficient in low-latency scenarios. However, Vegas can still deliver performance gains at a batch size of 8 for 10K contexts and at a batch size of 1 for 100K contexts (see Q3 of Reviewer MbP4). We will add this discussion in the paper.
> > >
> > >
> > > # Q3: Overfitting
> > > In our paper, the algorithm learns only from the last accepted token and may select tokens that are only critical to that token (“fitting to noise”). This makes the algorithm’s prediction of future critical tokens inaccurate (“hinders generalization”).
> > >
> > > We agree that, in the ML community, overfitting has a more specific meaning. We will replace the use of “overfitting” with the above explanation throughout the paper.

---

### Decision · Program_Chairs · 2026-04-30

**Decision:**

Accept (regular)

**Comment:**

This paper proposes Vegas, a method for accelerating long-context LLM inference via self-speculative decoding. The key motivation is that, in long-context settings, KV-cache read/write becomes a major bottleneck. Prior sparse-attention speculative decoding methods can speed up the drafting stage, but they typically rely on a separate KV-selection mechanism, creating a clear trade-off between drafting accuracy and selection overhead. The central insight of Vegas is to reuse the full-attention computation already available in the verification stage to guide sparse attention in subsequent drafting steps. This improves the acceptance of later draft tokens while avoiding the need for a separately designed and potentially expensive KV selector. In addition, the paper proposes Collect-2-Query, which gathers logits only from the first draft token and the bonus token, substantially reducing collection overhead.
Experimentally, the method is evaluated on models including Qwen3 and gpt-oss-20b, across benchmarks such as AIME25, CodeElo, and LongBench-v2. The paper reports 1.25×–2.81× throughput gains over vLLM, as well as an additional 1.18×–1.29× improvement over SoTA baselines. A series of ablation studies further supports the effectiveness and generalizability of the proposed design.

In sum, the paper presents a clear, technically sound, and practically useful solution to an important and realistic problem. The empirical evaluation and ablation studies are fairly comprehensive, and most reviewers (JwrV, sgBq, MbP4) moved to an explicitly positive stance after the rebuttal. Although one reviewer (V1N7) remains strongly opposed, that disagreement is mainly about the desired level of theoretical justification rather than about a concrete technical defect. I therefore recommend acceptance.